# Seeding Density and Nitrogen Fertilization Effects on Agronomic Responses of Some Hybrid Barley Lines in a Mediterranean Environment

**Giovanni Preiti ***, **Antonio Calvi, Maurizio Romeo, Giuseppe Badagliacca** and **Monica Bacchi**

Department AGRARIA, Mediterranea University of Reggio Calabria, 89122 Reggio Calabria, Italy;
antonio.calvi@unirc.it (A.C.); maurizio.romeo@unirc.it (M.R.); giuseppe.badagliacca@unirc.it (G.B.);
mbacchi@unirc.it (M.B.)
* Correspondence: giovanni.preiti@unirc.it; Tel.: +39-0965-1694274

**Abstract:** Over two cropping seasons, 2017/18 and 2018/19, an experimental trial was conducted in a typical cereal-growing environment of the Calabrian hills (southern Italy) to study seeding rate (D) and nitrogen fertilization (N) effects on two barley F1 hybrids (Zoo and Jallon) compared to those of a traditional variety (Lutece), assessing the bio-agronomic response. Barley hybrids, gradually introduced into the principal European countries starting in 2010 as winter forage, currently represent a significant part of the EU internal market. Productive performance was evaluated as grain yield for feed and total biomass for silage and/or biogas production. Research results pointed out the greater performance of barley hybrids compared to conventional varieties in terms of both grain and biomass production. On average, barley hybrids vigour mainly manifested itself through a high tillering and a greater number of ears m$^{-2}$ compared to those of the conventional variety (+24 and +23%, respectively). Furthermore, barley hybrids were characterized by a greater 1000-kernel weight and hectolitre weight than those of the Lutece variety (conventional variety). A significant increase in grain production was observed, increasing density from $D_{150}$ to $D_{225}$ rates (+35% and +33%, respectively) which was followed by a decrease in production shifting from $D_{225}$ to $D_{300}$ doses. A significant increase in biomass production was as well highlighted for the two hybrids, shifting from $D_{150}$ to $D_{225}$ rates (+26% and +27%, respectively). The applied nitrogen dose highlighted a different behaviour between the hybrids and the conventional variety; in particular, the lowest nitrogen dose ($N_{80}$) negatively influenced the Lutece variety both in terms of grain and biomass production (−9% and −16%, respectively) while the hybrids showed the best agronomic response even at the lowest dose. On average, with the $N_{80}$ dose, grain yield of Zoo and Jallon was greater than 20% and 16%, while with the $N_{120}$ dose grain yield was 9% and 7%, respectively. A similar behaviour was found for biomass yield. It should therefore be emphasized that barley hybrids possess high yielding capacities and that such higher grain production can be achieved in a Mediterranean environment by using a lower seed rate (approximately −25%) and a reduced nitrogen dose (approximately −33%) compared with those commonly applied to conventional varieties.

**Keywords:** barley hybrids; Zoo; Jallon; Lutece; seeding rate; nitrogen dose; sustainability

## 1. Introduction

Barley is a widely grown cereal throughout the world, with an estimated growing area of 44 Mha, production of 141 Mt [1], and domestication dating back to approximately 8000 B.C. in the Fertile Crescent [2]. It is an important crop, especially in areas where wheat cultivation is not feasible, thanks to some of its characteristics such as greater environmental adaptability, lower water requirements, tolerance to stress conditions, and ability to grow in either saline [3] and alkaline soils [4]. Barley has several applications, e.g., as seed in the malting industry and as livestock feed as a whole plant for hay, silage, green feed, or even as straw [5]. Hybrid breeding attempts in autogamous cereals, although moderately

successful, are mainly stimulated by the demand for higher productions in the evolving environmental situation dictated by abiotic stresses due to climate change [6]. Under stressful environmental conditions, barley landraces can give 25–61% higher yields than genetically improved varieties, whereas the latter give 6–18% higher yields under optimal conditions in West Asia [7]. Barley hybrids can benefit farmers with stable and higher yields than those of inbred lines, with an increasingly important market segment in Europe [8], because of maximum exploitation of heterosis [9], for example, in terms of traits related to higher biomass yields [10]. Plants with different genetic traits are crossed to obtain hybrids, in contrast to conventional varieties that are obtained from a single parental line through self-pollination [6]. As result of this hybrid process, vigorous plants with higher yields and superior agronomic traits are generated; in general, hybrids present a higher tillering index than do conventional varieties and undergo strong vegetative development by increasing root development, staying green, and showing greater resistance to adverse conditions [9]. In the case of self-pollinating crops, such as barley, producing hybrid seed is a major challenge as cross-pollination must be promoted and autogamous pollination must be prevented [11]. From the technique based on cytoplasmatic male sterility [12], hybrid seed can now be obtained by means of more efficient technologies, resulting in lower costs and a higher level of heterosis [13].

In general, the use of barley hybrids allows for savings on sowing seed quantity, ensuring a good seedbed preparation and avoiding late sowing with a recovery of the investment later between emergence and the stem elongation phase, while nitrogen inputs are still essential but lower than those used for conventional varieties [14]. Nitrogen plays a vital role in increasing crop yield. The application of the proper amount of nitrogen is the key to obtaining a better crop of barley with a good protein content. Nitrogen is an essential element for plants, and appropriate doses are required for strong plant growth and vigorous vegetative growth. Nitrogen stimulates tillering, possibly due to its effect on cytokine synthesis, and barley reacts to early N application by producing more tillers per plant and by exhibiting a higher tillers' survival rate. Careful monitoring of fertilizer application, especially nitrogen, is effective in preventing lodging [15]. Proper fertilisation programmes need to be developed, with nitrogen being applied at different times, taking into account environmental conditions and soil type [16]. Fertilisation management in fact affects not only the crop but also has an environmental impact, as part of the nitrogen that is distributed is lost to the ecosystem and becomes a source of pollution [17]. As nitrogen is one of the key factors influencing crop yields, the introduction of high-performance hybrids can improve yields and nitrogen use efficiency (NUE), which for barley is estimated to have increased by about 26% within a century of breeding activity [18]. Pankaj et al. (2015) [19] reported that nitrogen content in barley grain was influenced by the sowing date, while no significant influence was found for nitrogen content in straw regarding sowing date, variety, or nitrogen levels (maximum N content 0.35%). As mentioned above, due to its resistance to low water resources, barley is a widespread crop in the Mediterranean basin, which is characterized by variable rainfall and a semi-arid climate [20]. In Calabria (region of southern Italy), barley is grown on 8004 ha for a total production of 21,805 t [21]. Barley harvest residues, i.e., straw, can be a suitable substrate for bioenergy production through an anaerobic digestion process and is a low-cost sustainable material for green building. From an agronomic point of view, it is a common technique to bury the residues into the soil, preserving its carbon stock [1]; nevertheless, consideration should be given to the possibility of converting such agricultural waste into sustainable alternative uses. After grain harvest, the remaining straw in the field can be incinerated, and it is estimated that more than 20 million tons undergo this process [22], resulting in the production of greenhouse gases and loss of a potential resource that could be integrated back into the soil in a different form, for example, as digestate, i.e., organic fertilizer derived from a biogas plant [23]. As stated in Bernhard et al. [24], the methane yield per kg of dry matter for barley was comparable to that of maize, the most widely used cereal for biogas production [25], albeit with a lower DMY (dry matter yield), a superable shortcoming with sowing barley

hybrids [24], that generally develop larger culms with a higher number of tillers [1]. In a comparison of different biomasses for bioenergy production, Dinuccio et al. [26] reported that barley straw yielded about 15% higher methane than that of rice straw, although without a statistically significant difference. Silage is a common practice, which consists of a biochemical transformation pattern carried out by bacteria under anaerobic conditions [27] and used both to store straw for animal feed and for energy production [28]. Barley straw is characterized by a high TS (total solids) but a low sugar content necessary for the biochemical processes that occur during ensiling. To overcome this issue, Feng et al. [29] proposed cover crops and barley straw mixtures for ensiling for biogas production. In addition, the compositional variability of straw in relation to location, season, etc. must be considered [30].

In Italy, there are already many livestock farms that use animal manure mixed with various plant residues from cereal crops or from the agro-food industry as a renewable energy source through conversion within biogas plants. In light of the above, the aim of this work was to evaluate the bio-agronomic and productive behaviour of two barley hybrids, Zoo and Jallon, released by Syngenta through the Hyvido® technology, in relation to particular aspects of the cultivation technique. In this paper, the effect of different seeding rates and nitrogen fertilization strategies were evaluated on the productivity of barley hybrids compared to that of conventional varieties. In particular, the productive performance was evaluated both for grain production for livestock use and for total biomass production for silage and biogas. What follows represents a preliminary study on quantitative and qualitative characteristics of the above-mentioned hybrids in the Mediterranean environment, the results of which could be useful for further studies focused not only on production for the livestock sector (grain and/or silage) but also for alternative uses (biogas and bioconstruction).

## 2. Materials and Methods

### 2.1. Study Site

The experiment was carried out on a silt-sandy-loam soil (50.4% silt, 33.0% sand, 15.7% clay), classified as Fluventic Haploxerepts, coarse silty, mixed, and thermic [31], at the agricultural experimental centre of the Regional Agency for Agriculture "ARSAC" located in San Marco Argentano (Italy) (39°38′ N, 16°13′ E, 200 m asl) over two cropping seasons (2017/18–2018/19). The physical and chemical analyses of the soil in both seasons are shown in Table 1. The annual rainfall in San Marco Argentano was about 700 mm and the mean annual air temperature was 16.1 °C (1994–2019 period).

### 2.2. Experimental Design and Crop Management

A two-year experiment was conducted to determine the effects of seeding rate and nitrogen dose on the bio-agronomic behaviour of a six-row barley variety (Lutece), whose adaptability and stability of production are already widely validated in the test environment, and two six-row F1 hybrids (Zoo and Jallon cultivars—Hyvido® line Syngenta). Principal morpho-biological characteristics as well as the prevalent utilization of the three barley cultivars are reported in Table S1 (reported in "Supplementary material").

Seeding rate (D—150, 225, and 300 seeds $m^{-2}$), nitrogen rate (N—80 and 120 kg $ha^{-1}$) and three barley cultivars were compared based on a split-split-plot experimental design with three replications. Generally, a density of 300 germinable seeds $m^{-2}$ and a nitrogen dose equal to 120 kg $ha^{-1}$ are utilized for barley cultivated in the Mediterranean environment.

Nitrogen fertilization was differentiated only in the top dressing; in particular, 36 kg $ha^{-1}$ of N and 92 kg $ha^{-1}$ of $P_2O_5$ were applied as diammonium phosphate (200 kg $ha^{-1}$) before sowing, while another fertilization was carried out at the end of the tillering stage with urea using 44 kg $ha^{-1}$ ($N_{80}$) and 84 kg $ha^{-1}$ ($N_{120}$).

**Table 1.** Physical and chemical analysis of the experimental soil.

| Parameters | Seasons | |
|---|---|---|
| | 2017/18 | 2018/19 |
| Total sand (%) | 33.9 | 42.3 |
| Silt (%) | 50.4 | 44.7 |
| Clay (%) | 15.7 | 13.0 |
| Texture | Silt loam | Loam |
| $EC_{1:2}$ (d $Sm^{-1}$) | 0.141 | 0.206 |
| Soluble salts (g $kg^{-1}$) | 0.18 | 0.26 |
| $pH_{(KCl)}$ | 7.6 | 7.5 |
| Total $CaCO_3$ (g $kg^{-1}$) | 10.0 | 9.6 |
| Active $CaCO_3$ (g $kg^{-1}$) | 6.5 | 6.4 |
| $C_{org}$ (g $kg^{-1}$) | 12.1 | 12.4 |
| $N_t$ (g $kg^{-1}$) | 0.80 | 0.90 |
| C/N | 8.90 | 8.2 |
| Assimilable $P_2O_5$ (ppm) | 42 | 25 |
| Exchangeable $K_2O$ (ppm) | 105 | 107 |
| Exchangeable CaO (ppm) | 3473 | 3030 |
| Exchangeable MgO (ppm) | 282 | 243 |
| Exchangeable Na (ppm) | 10 | 27 |
| Mg/K (m.eq./100 g) | 6.30 | 5.30 |
| CEC (m.eq./100 g) | 23.4 | 20.8 |
| GSB | 60 | 59 |
| ESP | 0.18 | 0.56 |
| SAR | 0.02 | 0.05 |

The soil, previously cultivated with durum wheat in both years, was prepared for sowing by ploughing in summer to a depth of 30 cm, followed in autumn by two harrowing processes. Sowing dates were 18 November 2017 and 26 November 2018.

Sowing was carried out by plot seed drill (*Vignoli*) during the second half of November, in plots of 10 $m^2$ (1.44 × 7.00 m) in rows of 18 cm apart. Weed control was performed in mid-March using a mixture of pinoxaden, clopiralid, florasulam (Axial 60), and fluroxipir meptil (Columbus). Harvesting was carried out with a plot combine harvester "*Wintersteiger*" when the barley was fully ripe (stages 93–97).

### 2.3. Field and Yield Measurements

During the biological cycle of barley, some bio-morphological characteristics were detected. In the early phase of growth, earing date (stage 59) and plant height (stages 85–87) were determined as described in the BBCH scale [32]. Before harvesting, the number of plants per 1 $m^2$ area was counted to assess density. The biomass yield was determined a few days before harvest by collecting 0.36 $m^2$ (0.36 × 1 m), as a sampling area, for each plot, cutting about 5 cm above ground level. Furthermore 20 culms were collected for detailed measurements, specifically number and weight of grains per ears and thousand kernel weight.

After harvesting, grains from the 10 $m^2$ area, were weighed. On a sample of plot production, hectolitre weight and grain moisture were determined by carrying out three measurements using a special instrument (GAC II—Grain Analyzer Computer). The final grain yields per plot were converted into a standard moisture of 13%. For that reason, grain (100 g) and straw (100 g) samples were dried at 65 °C using a forced-air oven until a constant weight was reached. Dried samples were ground with a Wiley mill, subsequently passed through a 1 mm sieve, and analysed to determine total N (Kjeldahl method) and to calculate crude protein concentration. These data were multiplied with respective dry matter (DM) to obtain barley protein yield (AOAC) [33].

### 2.4. Statistical Analyses

Data were subjected to analysis of variance (ANOVA) using the GLM univariate procedure of IBM SPSS Advanced Statistics Version 22. The ANOVA was performed by using a mixed model analysis. Seeding rate, nitrogen rate, and cultivar were considered fixed effects. The year factor and its interactions with fixed factors were considered random effects, as were replicates within years.

Yang (2010) [34] suggested that in breeding and agronomic studies it may be more appropriate to consider year and location effects and their interactions with fixed effects as random, since the aim of most crop improvement programmes is to infer future performance at many untested locations.

The statistical significance of the effect was analysed using F-tests, whereas the differences between means were tested using the Tukey's Test HSD at $p \leq 0.05$ significance. Regression analysis was used to study the relationship between cropping density and some dependent variables, grain yield and biomass yield.

## 3. Results

### 3.1. Climatic Data

The observed environmental growing conditions during the experiment, typical of the Mediterranean climate, are shown in Table 2 together with the average values for a twenty-five year period (1994–2019).

**Table 2.** Monthly total rainfall and air temperature means (maximum and minimum) during the two growing seasons and long-term average (1994–2019).

| Month | Total Rainfall (mm) | | | Min Temperature (°C) | | | Max Temperature (°C) | | |
|---|---|---|---|---|---|---|---|---|---|
| | 2017/18 | 2018/19 | 25-Year | 2017/18 | 2018/19 | 25-Year | 2017/18 | 2018/19 | 25-Year |
| October | 181.0 | 13.0 | 72.6 | 11.8 | 13.7 | 11.0 | 22.9 | 21.9 | 24.2 |
| November | 122.4 | 100.8 | 102.9 | 7.8 | 9.7 | 7.3 | 16.9 | 17.7 | 18.8 |
| December | 47.2 | 63.6 | 88.9 | 5.1 | 5.7 | 4.6 | 13.0 | 13.9 | 14.9 |
| January | 81.4 | 30.6 | 88,5 | 6.2 | 2.1 | 3.5 | 14.0 | 10.3 | 13.8 |
| February | 32.2 | 124.8 | 72.7 | 4.7 | 4.8 | 3.8 | 12.1 | 13.5 | 14.6 |
| March | 81.2 | 106.6 | 77.9 | 7.6 | 7.6 | 5.1 | 15.5 | 16.9 | 16.8 |
| April | 56.8 | 4.8 | 51.8 | 10.3 | 9.0 | 7.4 | 21.5 | 18.2 | 20.1 |
| May | 86.0 | 40.8 | 34.0 | 13.9 | 10.8 | 11.0 | 24.4 | 16.8 | 25.2 |
| June | 11.6 | 77.4 | 21.8 | 16.9 | 17.2 | 17.2 | 28.0 | 34.3 | 31.2 |

The rainfall distribution during the two growing seasons showed high variability, most of it occurring in autumn and winter. Total rainfall (period October–June) was equal to 700 and 562 mm in the first and second growing seasons, respectively, in line with 611 mm recorded as the 25-year mean of the experimental site. During the first growing season, 43% of the total precipitation fell between October and November before sowing, whereas in the second growing season, February and March were the rainiest months. The decadal thermo-pluviometric trend in the period from October to June in the two years of the experiment is reported in Figure S1.

Air temperature regimes were similar for the two years and in line with the 25-year average of the trial site. The average monthly maximum temperatures during harvest season (October–June) were 18.7 °C and 18.2 °C, respectively. The average monthly minimum temperatures were 9.4 °C and 9.0 °C in 2017/2018 and 2018/2019, respectively. February 2016 was colder than February 2015 and the daily minimum air temperature dropped to 4.7 °C. In the first year, in line with the multi-year averages; from March, the air temperature started to increase steadily. In the second year, exceptionally, maximum temperatures remained significantly below seasonal averages throughout May.

The *p*-values of ANOVA for fixed factors and their interactions are reported in Table 3 along with error variance and variance associated to the effects of the interaction year × treatment (fixed effects), represented as percentage of the sum of the total

variance associated with the effect of the year. All the interactions between fixed factors and "year" random effect, except for the number of ears $m^{-2}$ and the tillering index, were not significant. Seeding density and nitrogen rate factors influenced the bio-agronomic behaviour of the barley cultivars under study differently.

**Table 3.** *p*-values—significant effects ($p \leq 0.05$) indicated in bold—of the analysis of variance for the fixed effects of seeding density (D), nitrogen rate (N), and barley variety (V) and their interactions on bio-agronomic and qualitative variables.

| Agronomic Traits | D | N | D × N | V | D × V | N × V | D × N × V | Error [1] | Y [2] Interaction |
|---|---|---|---|---|---|---|---|---|---|
| DF [3] | 2 | 1 | 2 | 2 | 4 | 2 | 4 | 72 | |
| Grain yield (t ha$^{-1}$) | **0.003** | 0.423 | 0.086 | **0.010** | **0.048** | 0.702 | **0.042** | 0.340 | 11.3 |
| Biomass yield (t ha$^{-1}$) | **0.003** | 0.458 | 0.147 | **0.003** | **0.022** | 0.910 | **0.031** | 0.270 | 13.0 |
| Earing (gg $\frac{1}{4}$) | 0.188 | 0.500 | 0.617 | **0.016** | 0.662 | 0.500 | 0.896 | 0.306 | 19.4 |
| Plant height (cm) | **0.048** | 0.716 | 0.336 | 0.283 | 0.940 | 0.623 | 0.877 | 11.027 | 18.2 |
| Ear density (n m$^{-2}$) | **0.006** | **0.043** | 0.931 | 0.052 | 0.549 | 0.451 | 0.661 | 51.525 | 6.5 |
| Tillering index | **0.045** | **0.016** | 0.281 | **0.047** | **0.044** | 0.279 | 0.320 | 0.929 | 16.6 |
| 1000-kernel weight (g) | **0.004** | 0.489 | 0.118 | **0.001** | 0.200 | **0.040** | **0.014** | 1.250 | 2.6 |
| Hectolitre weight (kg hl$^{-1}$) | **0.012** | 0.338 | **0.019** | **0.002** | 0.377 | **0.026** | **0.013** | 0.015 | 3.8 |
| Yield per ear (g) | 0.060 | **0.033** | **0.015** | **0.004** | **0.015** | 0.159 | 0.199 | 0.014 | 15.6 |
| Number of grains per ear (n) | 0.278 | **0.046** | 0.632 | 0.452 | **0.045** | 0.652 | 0.765 | 0.141 | 5.9 |
| Grain protein (g kg$^{-1}$) | **0.049** | 0.135 | **0.043** | 0.753 | 0.929 | 0.915 | 0.757 | 0.704 | 4.6 |
| Straw protein (g kg$^{-1}$) | **0.018** | 0.666 | 0.796 | 0.983 | 0.975 | 0.587 | 0.270 | 0.165 | 6.4 |
| Protein (t ha$^{-1}$) | **0.005** | 0.967 | 0.499 | **0.035** | **0.035** | 0.175 | 0.105 | 0.14 | 4.2 |

[1] Error variance. [2] Variance associated with the effects of the Year by treatment (fixed effects) interaction expressed as a percentage of the sum of the total variance associated with the effect of Year. [3] DF—degree of freedom.

During the two years of the experiment, the results highlighted the good productive potential of barley hybrids; their average production was very high both for grain and total biomass production. Seed germinability and physical purity of 95% and 98%, respectively, and the regular emergence in both years of the experiment insured the planned plant density. In addition, rainfall registered during the second decade of May in 2017/18 (49 mm) and the second decade of June in 2018/19 (63 mm) near the harvesting period did not cause any evident lodging phenomena, even with the higher nitrogen dose ($N_{120}$).

*3.2. Effect of Seeding Rate*

Seeding density significantly affected various aspects of barley productivity (Table 4). The barley cultivated with the highest seed density (300 germinable seeds $m^{-2}$) registered the highest grain and biomass production. Nevertheless, a significant increase in production should be noted shifting from $D_{150}$ (150 germinable seeds $m^{-2}$) to $D_{225}$ (225 germinable seeds $m^{-2}$); less important was the increase with the greater density, especially regarding grain yield (Figure S2).

Figure 1 highlights the relation between crop density, grain yield, and total biomass for barley hybrids and the traditional variety. All genotypes under study manifested a clear relationship between crop density, grain yield, and biomass. However, hybrid cultivars, in contrast to the conventional variety, showed a lower sensitivity to the investment increase, which appears clearly more pronounced for the grain productivity as demonstrated by the slope of the straight line.

The regression analysis is highly significant ($p \leq 0.01$); in addition, except for the Lutece variety, referring to the total biomass production (88%), more than 93% of the variability of the Y results are explained by the linear association with the X variable for both the investigated characteristics.

Total biomass quantity was the highest with $D_{225}$ and $D_{300}$ and their increases in total protein content were quite similar (0.79 and 0.80 t ha$^{-1}$, respectively).

Furthermore, the density rate increase ($D_{150}$ vs. $D_{300}$) influenced the crop bio-morphological behaviour, determining an early onset of three days on the earing phase, an increase in plant height of 5 cm, and an obviously greater number of spikes $m^{-2}$ that was, on average, between 354 and 503. In contrast, the lowest seeding density (150 germinable seeds $m^{-2}$) determined a significant increase on the tillering index (2.36), the 1000-kernel weight

(44.2 g), the hectolitre weight (66.8 kg hl$^{-1}$), and more generally on kernels production and number per spike (1.60 g and 50.5 n, respectively). The protein content of grain and straw resulted in similar values between the lowest (D$_{150}$ and D$_{225}$) and the highest densities (D$_{225}$ and D$_{300}$).

**Table 4.** Bio-agronomic behaviour of the genotypes under experiment in relation to treatments (fixed effects).

| Agronomic Traits | Seeding Rate | | | Nitrogen Dose | | Variety | | |
|---|---|---|---|---|---|---|---|---|
| | D$_{150}$ | D$_{225}$ | D$_{300}$ | N$_{80}$ | N$_{120}$ | Zoo | Jallon | Lutece |
| Grain yield (t ha$^{-1}$) | 4.87 c | 6.57 b | 6.72 a | 5.99 a | 6.12 a | 6.36 a | 6.23 b | 5.57 c |
| Biomass yield (t ha$^{-1}$) | 9.94 c | 12.40 b | 13.15 a | 11.28 a | 11.63 a | 12.50 a | 12.46 a | 9.40 b |
| Earing (gg $\frac{1}{4}$) | 23 a | 21 b | 21 c | 21 a | 22 a | 22 b | 24 a | 18 c |
| Plant height (cm) | 101 b | 105 a | 105 a | 103 a | 104 a | 103 b | 103 b | 105 a |
| Number of ears (n m$^{-2}$) | 354 c | 485 b | 503 a | 439 b | 456 a | 477 a | 479 a | 387 b |
| Tillering index | 2.36 a | 2.16 b | 1.68 c | 2.04 b | 2.10 b | 2.20 a | 2.21 a | 1.79 b |
| 1000-kernel weight (g) | 44.2 a | 42.2 b | 41.4 c | 42.7 a | 42.5 a | 43.0 a | 43.7 a | 41.1 c |
| Hectolitre weight (kg hL$^{-1}$) | 66.8 a | 64.6 b | 64.6 b | 65.5 a | 65.2 a | 66.1 a | 66.0 a | 64.0 b |
| Yield per ear (g) | 1.60 a | 1.48 b | 1.43 b | 1.51 a | 1.43 b | 1.46 b | 1.47 b | 1.58 a |
| Number of grains per ears (n) | 50.5 a | 46.3 b | 42.4 c | 46.9 a | 45.9 b | 45.3 b | 44.8 b | 49.1 a |
| Grain protein (g kg$^{-1}$) | 9.0 a | 8.7 ab | 8.4 b | 8.8 a | 8.7 a | 8.6 a | 8.8 a | 8.8 a |
| Straw protein (g kg$^{-1}$) | 2.6 a | 2.3 ab | 2.3 b | 2.4 a | 2.3 a | 2.4 a | 2.4 a | 2.4 a |
| Proteins (t ha$^{-1}$) | 0.63 b | 0.78 a | 0.78 a | 0.74 a | 0.72 a | 0.77 a | 0.77 a | 0.65 b |

For each treatment, different letters on the same row indicate significative differences according to Tukey's Test HSD at $p \leq 0.05$.

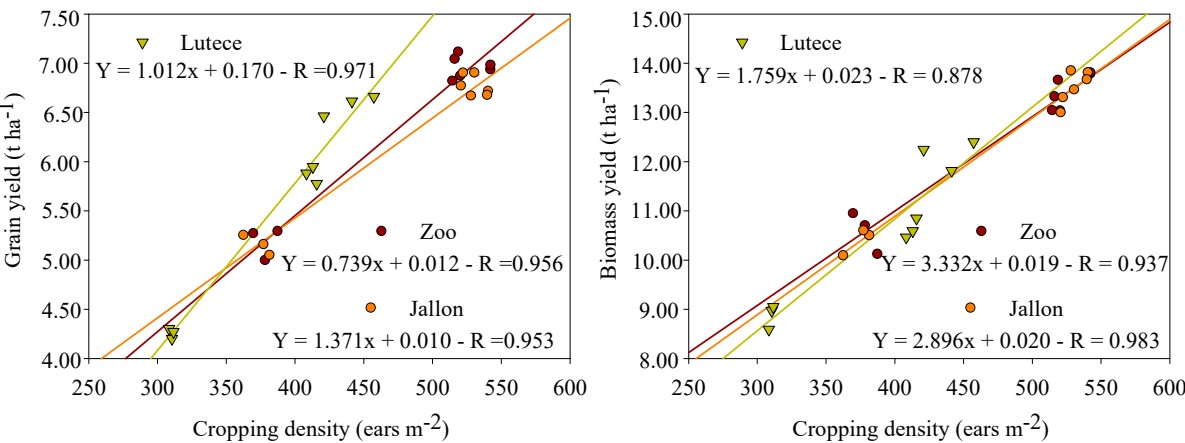

**Figure 1.** Relation between crop density, grain yield, and biomass yield for the conventional barley variety (Lutece) and for the barley hybrids (Zoo and Jallon).

*3.3. Effect of Nitrogen Dose*

The effect of nitrogen rate on the bio-agronomic behaviour of barley was more contained: grain and biomass yield values were quite similar. Overall, the barley that was fertilized with the highest nitrogen dose (120 kg ha$^{-1}$) showed an increased tillering index that consequently influenced spikes number m$^{-2}$ (+17), a significant increase. In contrast, the lower nitrogen dose (N$_{80}$) positively influenced both yield per ear (1.51 g) and the number of grains per spike (46.9). No significant differences were found in grain and straw protein content, producing quite similar results despite the different N rates (Table 4).

*3.4. Effect of Variety*

Variety influenced both the bio-agronomic and qualitative aspects of barley (Table 4). Hybrid varieties Zoo and Jallon showed productive levels, grain yield, and total biomass that were significantly higher than those of the conventional variety Lutece; in particular, the Zoo variety was the most productive with a grain yield of 6.36 t ha$^{-1}$, followed by Jallon

(6.23 t ha$^{-1}$) and Lutece (5.57 t ha$^{-1}$). Total biomass production of the hybrid varieties was clearly higher as well compared to that of Lutece (+18.2%, on average). Hybrid varieties differed from the conventional variety in longer biological cycle (+5 days, on average) and in lower plant height ($-2$ cm). The higher number of ears at harvest should also be noted for Zoo and Jallon (477 and 479 spikes m$^{-2}$, respectively) compared to Lutece (387 spikes m$^{-2}$). Barley hybrids showed a greater tillering capacity compared to that of the conventional variety (+23%). Moreover, they showed superior qualitative performance and greater 1000-kernel weight ($\geq$43.0 g), hectolitre weight ($\geq$66.0 kg hL$^{-1}$), and contribution to protein production (equal to 0.77 t ha$^{-1}$).

### 3.5. Interactions

The study of interactions between principal factors allowed us to deeply examine the information obtained from this research. The interaction "seeding density $\times$ nitrogen" was significant for only three of the traits detected: hectolitre weight, production per ear, and grain protein content (Figure 2).

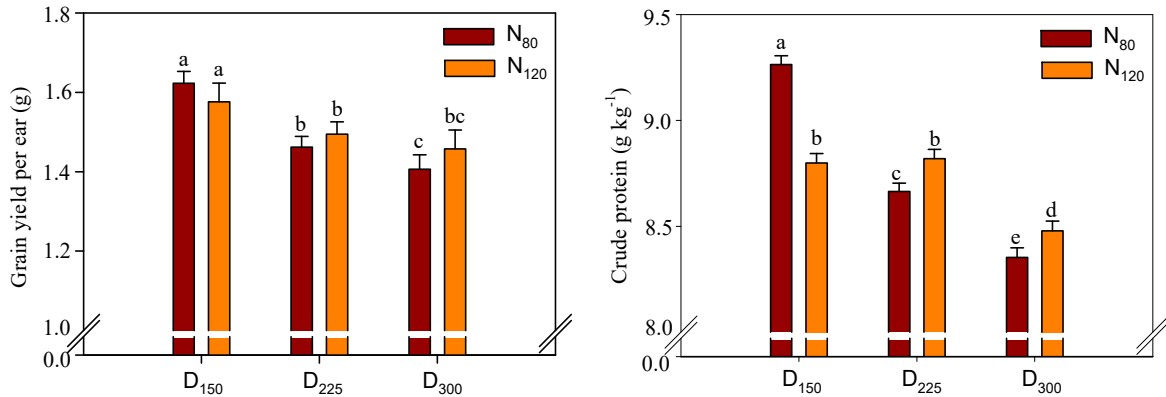

**Figure 2.** Grain yield per ear and crude protein content depending on seeding density (D) and nitrogen dose (N) (mean $\pm$ SD, $n$ = 18). Within each graph, bars marked with different letters are statistically different ($p \leq 0.05$) according to Tukey's HSD Test.

The lower seeding density was more effective in determining the degree of caryopsis filling regardless of the nitrogen dose distributed. Indeed, the highest hectolitre weight was identified with D$_{150}$ (+2.2 kg hL$^{-1}$) where no significant differences were detected between the two nitrogen levels, while in D$_{225}$ and in D$_{300}$ values were lower and varied according to nitrogen level (data not shown).

Production per ear and grain protein content were influenced by the interaction between these two factors as well (Figure 2). The highest production per ear was recorded with D$_{150}$ (1.60 g), followed by D$_{225}$ (1.48 g); in both cases the nitrogen dose did not lead to significant differences.

However, with the D$_{300}$ the recorded values were lower and varied according to the nitrogen level (1.46 and 1.41, respectively).

In addition, the grain protein content was influenced by these two factors. At D$_{150}$ it was higher with N$_{80}$ whereas in D$_{225}$ and D$_{300}$ the grain protein content was higher with N$_{120}$.

The interaction between seeding density and variety (Figure 3) was particularly useful for understanding the productive behaviour of the hybrid and conventional varieties in relation to the crop intensification.

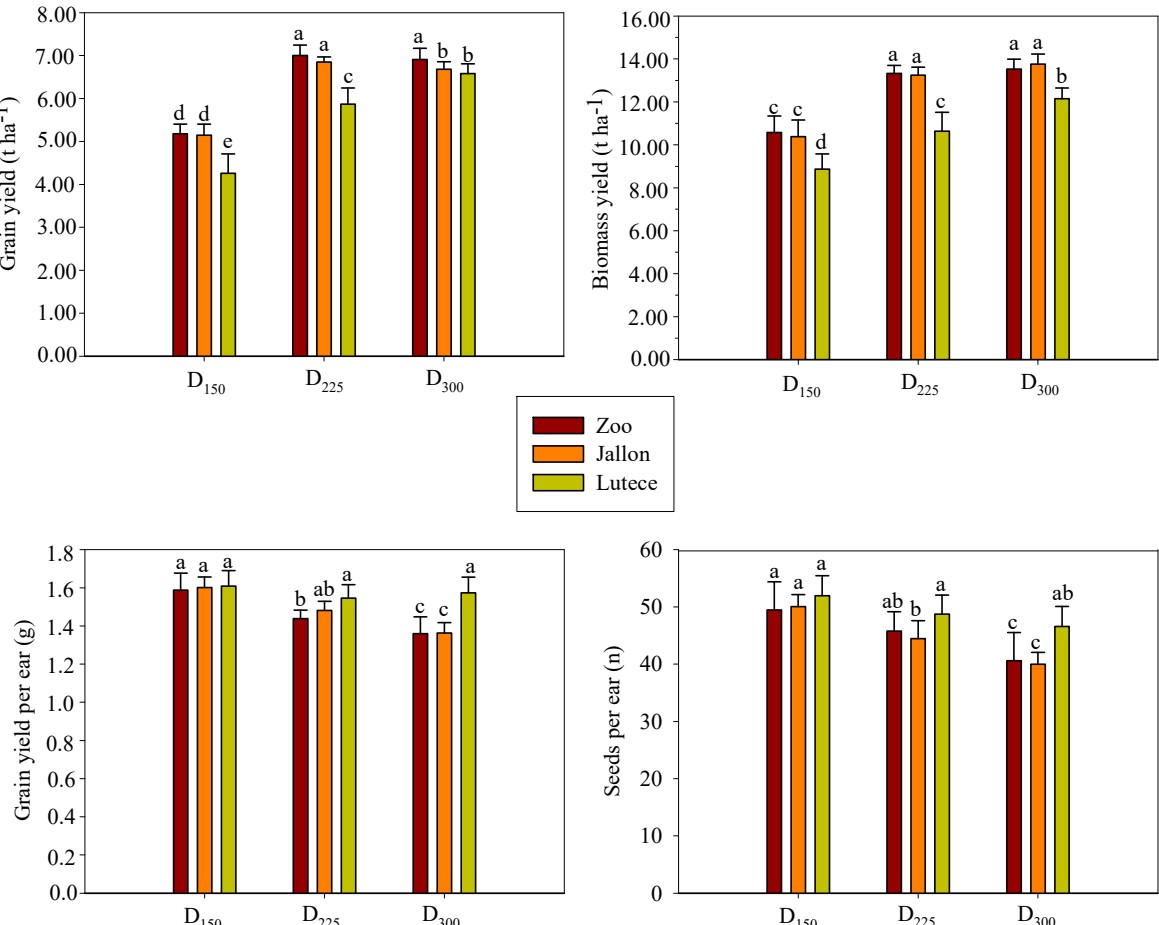

**Figure 3.** Grain yield, biomass yield, production per ear, and number of seeds per ear depending on seeding density (D) and variety (mean $\pm$ SD, *n* = 12). Within each graph, bars marked with different letters are statistically different ($p \leq 0.05$) according to Tukey's HSD Test.

Concerning grain yield, it should be noted that the hybrid varieties showed significant differences moving from $D_{150}$ to $D_{225}$ but not from the latter to $D_{300}$, while the conventional variety showed significant increases going from lower to higher seeding density.

The same trend was shown for total organic production. Regarding the yield components (yield and seed number per ear), the tested cultivars reacted differently to the increase in seeding density.

In particular, the conventional variety showed no significant difference from $D_{150}$ to $D_{300}$, while the hybrid cultivars showed a steady increase in both production and seed number per ear.

In addition, seeding density significantly affected both tillering index and protein yield per hectare, as expected (data not shown). Hybrid varieties showed a significantly higher tillering capacity than did the conventional variety, especially in the lower density ($D_{150}$).

Protein production per hectare was positively correlated both with grain yield and biomass yield (r = 0.798 \*\*\* and 0.617 \*\*\*, respectively).

Nitrogen fertilization had significant influence on the tested cultivars (Figure 4).

Overall, there was an increase in tillering index, which was reflected in a higher number of ears per $m^2$ at harvest; this increase was more evident in the conventional variety (+7%) than in the hybrids (+3%).

Nevertheless, the Lutece variety responded positively and significantly to $N_{120}$ (standard dose for the test environment) in terms of both grain yield and biomass production (+9% and +16%, respectively) while no advantage was recorded regarding the production of hybrid varieties.

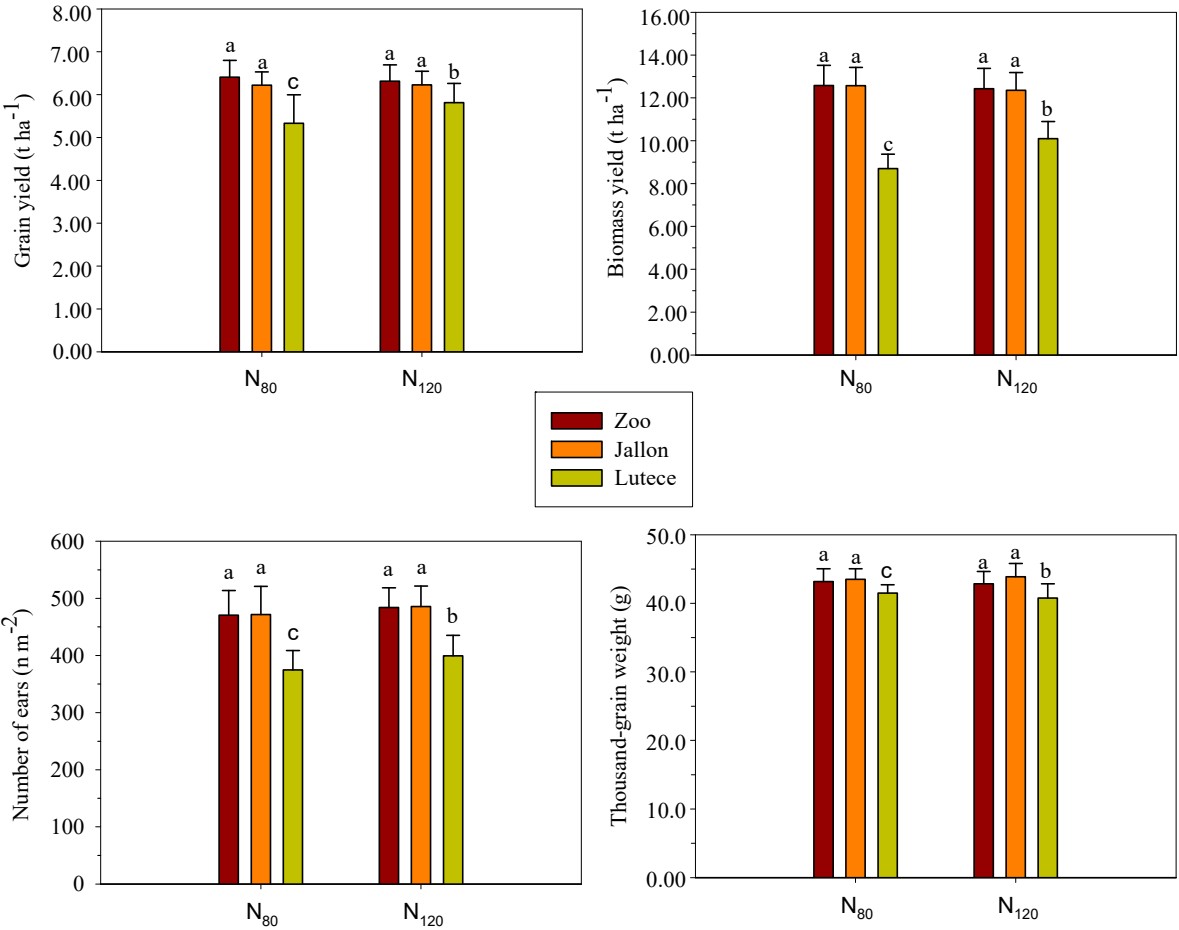

**Figure 4.** Grain yield, biomass yield, number of ears, and 1000-kernel weight depending on nitrogen dose (N) and variety (mean ± SD, *n* = 18). Within each graph, bars marked with different letters are statistically different (*p* ≤ 0.05) according to Tukey's HSD Test.

Furthermore, nitrogen rate affected the quality traits of the grain leading to a significant decrease in 1000-kernel weight and hectolitre weight from $N_{80}$ to $N_{120}$, while no significant changes were recorded for the hybrid varieties.

The interaction $D \times N \times V$ was also significant for some of the measured traits (grain yield, biomass production, 1000-kernel weight, and hectolitre weight), showing a different response of the cultivars compared to increases in both seeding density and nitrogen availability (Tables S2 and S3).

## 4. Discussion

Year effect was a significative source of variance for most of the examined parameters (data not shown). Bio-agronomic behaviour of the crop, albeit to a limited extent, was affected by precipitation, especially by the different rainfall distribution together with a differing temperature trend during the two experiment years. Nevertheless, the aim of the present study was to assess how seeding rate, nitrogen rate, and variety might influence barley hybrids' yield, excluding the environmental effects that would be documented later. Consistency of the effects of treatments with the year factor was a question; for this reason, we felt it appropriate to report the percentage of the environment variance with the sum of "year*treatment" interactions (Table 3). The overall percentage was limited (<20%); in particular, for earing time, plant height, 1000-kernel weight, hectolitre weight, grains number per ear, grain protein content, straw protein content, and yield proteins per hectare, percentage was low (1–10%), but it was higher for barley parameters such as grain production, biomass production, ears number $m^{-2}$, tillering index, and yield per ear (11–20%). Regarding the tillering index, results (Table 4) indicated a significant

decrease at higher seeding rates, in agreement with O'Donovan et al. [35]. Seeding and nitrogen rates are well known for traditional barley varieties production; nonetheless, the effects on productivity and quality of barley hybrids are not known with reference to the Mediterranean environment. Our results demonstrate the good productive potential and stability of hybrids barley in a semi-arid environment. The bio-agronomic performance of these barleys confirms what has already been observed by Mühleisen et al. [9] in other environments; barley hybrids are, in fact, characterized by a higher vegetative vigour and a longer stay green as well as a lower susceptibility to abiotic and biotic stresses. The barley hybrids vigour was mainly manifested in their high tillering capacity, which resulted in a higher average number of ears per $m^{-2}$ at harvest than that of the conventional variety (+23 and +24%, respectively), in 1000-kernel weight (+5 and +6%, respectively), and hectolitre weight (+3%). On the other hand, the Lutece variety showed, compared to the hybrids, the highest yield components in production per ear and in kernels number per ear.

The results obtained in this study further demonstrate that barley hybrids behave differently from the conventional variety when increasing the seeding density. For the two hybrids (Zoo and Jallon) there was a significant increase in grain yield from $D_{150}$ to $D_{225}$ doses (+35% and +33%, respectively) and a slight decrease in production from $D_{225}$ to $D_{300}$. In contrast, the conventional variety (Lutece) showed a continuous and significant increase in grain production switching from $D_{150}$ to $D_{225}$ and from this to $D_{300}$ (+37% and +12%, respectively). Regarding the total biomass production for the two hybrids, there was a significant increase from $D_{150}$ to $D_{225}$ (+26% and +27%, respectively) and a slight increase (+2% and +4%, respectively) from $D_{225}$ to $D_{300}$ doses. Overall, the increase in productivity of the hybrid varieties, Zoo and Jallon, in comparison to Lutece, was 14% and 12% for grain production and 19% and 18% for total biomass production, respectively. Therefore, it is recommended that barley hybrids be sown at a lower seeding rate (about −25%) than traditional varieties in a Mediterranean environment. These results confirm the improved yield performance obtained with $D_{225}$, in line with reports on yield stability of barley hybrids and other micro-thermal cereals in various German environments [9]. It should also be noted that the increase in seeding rate did not lead to a significant increase in the grain protein content, which was similar between the different seeding densities under study and in contrast with what has been reported in Edney et al. [36] on malting barley, in which higher levels of seeding rates corresponded to lower levels of grain protein while higher nitrogen rates corresponded to increased grain protein content. The nitrogen dose affected the production response of the checked varieties (hybrids and conventional varieties) in different ways. Although Terefe et al. [37] reported that an increase in nitrogen inputs within certain doses leads to a consequent increase in grain yields, in the present study, the hybrid varieties, differently from Lutece, did not exhibit an advantage in terms of production when increasing the nitrogen rate. This could be explained by the strong vegetative vigour and development of the root system, which allow the barley hybrids to search for and efficiently use the residual mineral and organic nitrogen that gradually becomes available in the soil in spring, as reported in Blandino et al. [14]. This would also show a high nitrogen use efficiency (NUE), considering that the increase in total protein per hectare (grain + straw) was about 16% for hybrid varieties, mainly related to their higher productivity. In conclusion, compared to conventional varieties under conditions of good fertility, it becomes convenient to reduce nitrogen inputs not only to contain costs, but also to avoid nitrogen losses through leaching and to prevent excessive vegetative vigour and the consequent increased lodging risk.

## 5. Conclusions

In the present research, the bio-agronomic behaviour of two F1 barley hybrids compared to a conventional variety in response to three different seed densities and two nitrogen levels was studied. Our results show that hybrids, on average, are most productive in terms of both grain and total biomass production (+13% and +18%, respectively) than the conventional variety, and that these results can be achieved by using about 25%

fewer seeds per m$^{-2}$. Moreover, hybrid varieties showed a better production response both in grain production and total biomass, reducing the nitrogen dose by 33% compared to that of traditional varieties.

In addition to increased grain production, the introduction of these hybrids in southern Italian cropping systems earns serious consideration for the harvesting of grains for silage production and/or for biogas production.

Finally, the reductions in seeds and nitrogen rate per hectare, when compared to those of the conventional variety, is advantageous both economically and in terms of environmental sustainability.

**Supplementary Materials:** The following are available online at https://www.mdpi.com/article/10.3390/agronomy11101942/s1, Figure S1—Decadal rainfall (mm), average maximum and minimum temperatures during the growing seasons. Figure S2—Grain yield (A) and biomass yield (B) in relation to seeding density; indicates the standard error of the mean. Table S1—Morphological and physiological characteristics and use of the genotypes under study. Table S2—Density × nitrogen × variety interaction per grain yield (t ha$^{-1}$) and biomass yield (t ha$^{-1}$). Table S3—Density × nitrogen × variety interaction per 1000-kernel weight and hectolitre weight.

**Author Contributions:** Conceptualization: G.P. and M.B.; methodology: G.P., M.B. and A.C.; Formal analysis: G.P. and G.B; investigation: G.P., A.C., M.R. and G.B.; writing—original draft: G.P., M.B. and A.C.; writing—review and editing: G.P. All authors have read and agreed to the published version of the manuscript.

**Funding:** This research received no external funding.

**Institutional Review Board Statement:** Not applicable.

**Informed Consent Statement:** Not applicable.

**Data Availability Statement:** Not applicable.

**Conflicts of Interest:** The authors declare no conflict of interest.

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
