# Peer review of "Seeding Density and Nitrogen Fertilization Effects on Agronomic Responses of Some Hybrid Barley Lines in a Mediterranean Environment"

_agronomy, doi:10.3390/agronomy11101942_

Round 1

Reviewer 1 Report

  1. Please use SI units.
  2. In the introduction, please limit the information on straw (L. 73-99) and extend it on nitrogen fertilization.
  3. Was mineral fertilization with N and P only used?
  4. Please add references to the all methods used.
  5. What was the difference between biomass yields (L. 155) and grains and straw (L. 160)?
  6. Please correct language and editing errors, e.g. Novemberre (Table 2).
  7. Please introduce every abbreviation and acronym before using it in the text (put them in parentheses after the full terms).
  8. All abbreviations and acronyms used in tables and figures should be defined in the table notes or figure captions.

Author Response

Dear Reviewer

Attached are the responses to your comments

  1. Please use SI units.

We thank the Reviewer for the suggestion. We have corrected the text according to SI units.

  1. In the introduction, please limit the information on straw (L. 73-99) and extend it on nitrogen fertilization.

We accepted the reviewer's suggestion by reducing the straw information and integrating the nitrogen information.

  1. Was mineral fertilization with N and P only used?

Yes, fertilization involved only nitrogen and phosphorus supply. In soils of southern Italy potassium availability is almost everywhere good. For this reason, potassium fertilization is not done except in soils particularly lacking of this element.

  1. Please add references to the all methods used.

We have included in the text the methodology used for hectolitre weight, moisture and protein content determination.

  1. What was the difference between biomass yields (L. 155) and grains and straw (L. 160)?

Thank Reviewer for the comment. We have incorrectly reported that after harvest, grain and straw weight was determined for each individual parcel. In reality, only the grain production per plot was weighed.

  1. Please correct language and editing errors, e.g. Novemberre (Table 2).

We have improved English language and style and corrected editing errors.

  1. Please introduce every abbreviation and acronym before using it in the text (put them in parentheses after the full terms).

Done.

  1. All abbreviations and acronyms used in tables and figures should be defined in the table notes or figure captions.

Done.

Reviewer 2 Report

Preiti et al. presented in the manuscript the results of studies on the cultivation of barley in Mediterranean environment conditions with various seeding densities and nitrogen fertilization. The results are of particular interest to farmers, but also provide important information for plant breeders. The manuscript was well prepared. My only remark is that the authors take into account the standard deviation in the graphs.

Author Response

Dear Reviewer

Attached are the responses to your comments

  1. My only remark is that the authors take into account the standard deviation in the graphs. 

Done, we have included the standard deviation in the graphs.

Round 2

Reviewer 1 Report

The authors have addressed the comments and suggestions I made in the review. I have no further suggestions.